# Quality of Refrigerated Squid Mantle Cut Treated with Mint Extract Subjected to High-Pressure Processing

**DOI:** 10.3390/foods13081264

**Published:** 2024-04-20

**Authors:** Krisana Nilsuwan, Suriya Palamae, Jasmin Naher, Natchaphol Buamard, Bin Zhang, Soottawat Benjakul

**Affiliations:** 1International Center of Excellence in Seafood Science and Innovation, Faculty of Agro-Industry, Prince of Songkla University, Songkla 90110, Thailand; krisana.n@psu.ac.th (K.N.); suriya.pal@psu.ac.th (S.P.); jasmin17hstu@gmail.com (J.N.); natchaphol.b@psu.ac.th (N.B.); 2Key Laboratory of Health Risk Factors for Seafood of Zhejiang Province, College of Food and Pharmacy, Zhejiang Ocean University, Zhoushan 316022, China; zhangbin@zjou.edu.cn; 3Department of Food and Nutrition, Kyung Hee University, Seoul 02447, Republic of Korea

**Keywords:** squid mantle cut, mint extract, high-pressure processing, next-generation sequencing, quality and safety, shelf-life

## Abstract

Squid (*Loligo vulgaris*) is commonly prone to spoilage, leading to a short shelf-life. High-pressure processing (HPP) can play a role in maintaining the quality and freshness of squid. Along with HPP, food preservatives from natural sources such as mint extract (ME), which are effective, safe, available, and cost-effective, are required. The present study aimed to investigate the combined effect of ME and HPP on the quality of refrigerated squid mantle cuts (SMC) over a period of 15 days. The time-kill profiles of ME and planktonic cell inactivation by HPP were assessed. ME (400 mg/L) inhibited bacterial growth, while planktonic cells treated with HPP (400 MPa) exhibited a reduction at 5 min. Physicochemical and microbial qualities of SMC treated with ME (0, 200, 400 mg/L) followed by HPP (0.1, 200, 400 MPa) for 5 min were monitored during refrigerated storage. Samples treated with ME (400 mg/L) and HPP (400 MPa) exhibited lower weight loss, cooking loss, pH changes, volatile base content, microbial counts, and higher textural properties than other samples. Based on next-generation sequencing results, *Brochothrix campestris* from family *Listeriaceae* was the predominant spoilage bacteria in treated sample after 12 days of storage. Therefore, ME and HPP combined treatments exhibited effectiveness in extending the shelf-life of refrigerated SMC.

## 1. Introduction

High-pressure processing (HPP) has emerged as a cutting-edge, non-thermal technology for the food industry. It can contribute to microbial reduction while its sensory and nutritional qualities remain uncompromised [1]. HPP is now widely used for the treatment of vegetables, fruit juices, and seafood [2,3]. Therefore, HPP could be an alternative processing technology to prolong the shelf-life of seafood [4]. Spoilage of seafood can be retarded by HPP, and HPP does not intensively affect the chemical composition of food [5]. However, seafood, including vertebrates, molluscs, and crustaceans are prone to microbial and chemical deterioration due to their high contents of polyunsaturated fatty acids (PUFAs) and non-nitrogenous compounds [6]. This leads to a short shelf-life. Among molluscs, squid is consumed as a popular seafood, which is commonly used in Thai cuisine [7]. These include stir-fried, grilled squids, curries and salads [8]. However, the post-harvest changes of squid cause economic loss and rejection by consumers. To overcome such drawbacks, plant extracts along with HPP could enhance the preservation of squid. However, squid’s rubbery texture makes it more difficult to chew. To tackle this problem, HPP could serve as a promising non-thermal technology to improve the tenderness of treated squid [9]. From other studies, it has been observed that 50% of the squid mantle structure was deformed when >600 MPa pressure was used, because high pressure may induce the destruction of muscle, where disruption and denaturation of native collagen and proteins occur [9]. In addition, HPP can play an important role in maintaining the quality and freshness of squid. The enzymatic activity in the squid mantle can lead to quality loss, thus negatively affecting odor, taste, color, and texture. However, HPP could effectively inactivate those enzymes [8].

Along with HPP, the use of natural preservatives has also drawn attention to reducing food loss by up to 40% each year, primarily attributed to inactivation of microorganisms encompassing bacteria, yeast, and molds [10]. To reduce these problems, chemical preservatives are applied to multiple food items to maintain their quality and retard microbial growth [11]. However, continuous use of these chemical additives has harmed human health [12]. As a consequence, food preservatives from natural sources that are effective, safe, available, and cost-effective are required [13]. Plant extracts rich in phytochemicals have been used as natural additives. These are classified into different groups based on their chemical structure, mode of action, and functional groups, consisting of phenolic compounds (terpenes/terpenoids, polyphenols, organic acids, and tannins) [10]. Phenolic compounds contain multiple OH^-^ groups present in their aromatic rings, either in monomeric (phenol) or polymeric forms (polyphenols), organic acids (COOH^−^), and tannins. These compounds can lead to a drastic change in the microflora structure and cause cell damage. In addition, phytochemicals were classified as GRAS (generally recognized as safe) due to their lower risk of developing bacterial resistance than their chemical counterparts [11]. Apart from antimicrobial activity, plant extracts show antioxidant activity to prevent lipid oxidation in the food system. Recently, Naher, et al. [14] reported that mint extract had both antimicrobial and antioxidant activities. It can be used to extend the shelf-life of seafood in conjunction with potential non-thermal processing technologies like HPP. 

Therefore, this study aimed to investigate the combined effect of mint extract (ME) and HPP on the quality changes of squid mantles cut during refrigerated storage. Furthermore, the microbial community in the squid mantle cut was also determined by next-generation sequencing (NGS).

## 2. Materials and Methods

### 2.1. Chemicals and Microbial Media

All analytical grade chemicals were purchased from Sigma-Aldrich (St. Louis, MO, USA). Plate count agar, *Pseudomonas* isolation agar, triple sugar iron, and eosin methylene blue were procured from Thermo Fischer Scientific (Waltham, MA, USA) and Hi Media Laboratories LLC (Mumbi, Maharashtra, India). 

### 2.2. Preparation of Mint Extract and Time-Kill Profile Study

Fresh mint (*Mentha arvensis*) leaves were collected from the fresh market in Hat Yai, Songkhla, Thailand. The leaves were then dried in an oven, blended, and sieved. The extraction was performed using 80% ethanol (*v*/*v*) as described by Naher, Nilsuwan, Palamae, Hong, Zhang, Osako, and Benjakul [14] and denoted as ME. 

The time-kill analysis of ME was conducted as tailored by Palamae, et al. [15]. *Pseudomonas aeruginosa* (ATCC 27834) and *Shewanella* spp. (TBRC 5775), representing the spoilage bacteria in seafood products, were cultured overnight (37 °C) and adjusted to ~10^7^ CFU/mL using the 0.5 McFarland standard. The bacterial suspension without extract was taken as a positive control. Subsequently, 11.80 mL of ME at 400 mg/L was then mixed with 120 μL of bacterial suspensions and placed in a continuous shaker at 37 °C. The sample (100 μL) was taken, and the number of colonies was counted at specific time intervals (0, 2, 6, 18, and 24 h) by using the spread plate method and expressed as log CFU/mL. A graph between the count and time (h) was plotted. 

### 2.3. Inactivation of Planktonic Cells by High-Pressure Processing

An 0.5 mL aliquot of *P. aeruginosa* (ATCC 27834) and *Shewanella* spp. (TBRC 5775) was aseptically transferred from the stock culture to 3 mL of sterile tryptic soy broth (TSB). For both species, the McFarland standard was used for dilution to achieve a final concentration of ~10^8^ CFU/mL. The planktonic cells were inactivated by subjecting the bacterial suspension to HPP at a fixed pressure level (400 MPa) for varying times (0, 1, 3, 5, and 7 min). Bacterial colonies were determined and expressed as log CFU/mL. The protocol to determine planktonic cell inactivation, as tailored by Mittal, et al. [16] was adopted. 

### 2.4. Preparation of Squid Mantle Cut Treated with Mint Extract Combined with HPP

To prepare squid mantle cut (SMC), the freshly caught squids (*Loligo vulguris*) with an average weight of 25.0 ± 0.5 g were procured (within 12 h of harvesting) from the fresh market and brought to the laboratory within 1 h, where a sample/ice ratio of 1:1 was maintained. Then, squids were beheaded, eviscerated, deskinned, and cleaned with a 3% brine solution. Subsequently, squid mantles were cut into small pieces (5 × 5 cm). The ME was prepared and applied to SMC to obtain the final concentrations of 0, 200, and 400 mg/L, as described by Naher, Nilsuwan, Palamae, Hong, Zhang, Osako, and Benjakul [14]. Finally, all treated samples were placed in a linear low-density polyethene-polyamide bag and vacuum-sealed with an Audionvac VM203 machine (Audiovac, Weesp, The Netherlands). 

For HPP treatment, the method of Palamae, et al. [17] was adopted. A high-pressure processing unit (Model HPP 600 MPa 5 L, Jiujin, Baotou KeFa High-Pressure Technology Co., Ltd., Baotou, China) with a capacity of 5 L was used, where sterile water served as the pressure-transmitting medium. Pressure levels were fixed at 200 and 400 MPa for 5 min. The vacuum-sealed pack was placed in a 5 L plastic container with a specified water flow rate. All samples were denoted as M0P2, M2P2, and M4P2 for SMC treated with ME at 0, 200, and 400 mg/L, followed by HPP of 200 MPa for 5 min, respectively. M0P4, M2P4, and M4P4 represented SMC treated with ME at 0, 200, and 400 mg/L and subjected to HPP at 400 MPa for 5 min, respectively. A sample without any treatments was used as the control. All samples were stored at 4 °C for 15 days and subjected to analyses at every 3-day interval.

### 2.5. Analyses

#### 2.5.1. Microbial Analysis

The evaluation of total viable count (TVC), psychrophilic bacteria count (PBC), *Pseudomonas* spp. count (PDC), H_2_S-producing bacteria count (HSPBC), and *Enterobacteriaceae* count (EBC) was conducted using the methodology outlined by Palamae, Temdee, Buatong, Suyapoh, Sornying, Tsai, and Benjakul [17]. The microbial count was expressed in log CFU/g.

For microbiological analysis, SMC (10 g) was collected aseptically in a sterilized stomacher bag and cut into small pieces. Subsequently, to obtain homogenous samples, 90 mL of 0.85% saline solution was blended with the sample in a stomacher blender (Stomacher M400, Seward Ltd., Worthington, UK) at 230 rpm for 2 min. Furthermore, 0.1 mL of aliquots was serially diluted and spread onto plate count agar to determine the TVC and PBC and incubated at 37 °C and 4 °C for 24 h and 7 days, respectively. Furthermore, the PDC and HSPBC were determined using *Pseudomonas* isolation agar and triple sugar iron, respectively, and incubated at 37 °C for 24–48 h where the colony color was blue-green or green on *Pseudomonas* isolation agar, while black colonies were detected as HSPB. Additionally, EBC was determined at 37 °C for 24–48 h on eosin methylene blue (EMB) agar.

#### 2.5.2. Chemical Analysis

Total Volatile Base (TVB) and Trimethylamine (TMA) Contents

TVB and TMA contents in the samples were determined according to the Conway microdiffusion method, as detailed by Temdee, et al. [18]. In brief, 2 g of sample was mixed with 8 mL of a 4% (*w*/*v*) TCA solution, homogenized at 11,000 rpm for 2 min, and the homogenate was left at room temperature for 30 min. Filtration was accomplished through a Whatman No. 41 filter paper (Whatman International, Ltd., Maidstone, England). The collected filtrate was subjected to analysis using the Conway unit. Blanks were also prepared for each analysis by adding 4% TCA instead of a sample. TVB and TMA contents were expressed as mg N/100 g of sample. 

pH Value

The pH measurement was performed as described by Nirmal and Benjakul [19]. Sample (2 g) was mixed with 20 mL of distilled water and homogenized at a speed of 11,000 rpm for 3 min (IKA Labortechnik, Selangor, Malaysia). Afterwards, the pH was measured at room temperature using a pH meter (Eutech Instrument-pH 700, Eutech Instrument Singapore). 

Peroxide Value (PV) and Thiobarbituric Acid Reactive Substances (TBARS)

PV and TBARS value were determined as explained by Nirmal and Benjakul [19]. Cumene hydroperoxide in the range of 0.2–0.6 M was used as a standard curve. PV was expressed as mg cumene hydroperoxide/kg sample after blank subtraction. 

For TBARS determination, the procedure of Temdee, Singh, and Benjakul [18] was adopted. Malondialdehyde (MDA) (60–600 μM) was used for standard curve preparation. The value was reported as mg of MDA equivalent/kg of sample. 

#### 2.5.3. Weight Loss and Cooking Loss

The weight loss and cooking loss of a sample were measured as per the method of Temdee, Singh, and Benjakul [18]. For the weight loss, the initial weight of the sample before storage and the final weight of the sample at the designed storage time were examined and used for calculation. 

For cooking loss, the initial weight of the sample was determined. Thereafter, all samples were steamed for 5 min until the core temperature of the samples reached 85 °C, then cooled in iced water for 2 min, and drained at 4 °C for 5 min. The final weight of the steamed sample was measured, and cooking loss was calculated.

#### 2.5.4. Texture Analysis

Firmness and toughness were determined as described by Temdee, Singh, and Benjakul [18]. The measurement was accomplished by a texture analyzer (Model TA-XTplus, Serial 12471, Stable Micro Systems, Surrey, UK) equipped with a Warner-Bratzler blade with a crosshead speed of 2 mm/s, 25 mm distance, and a 50 kg load cell. The shear force perpendicular to the axis of each mantle cut was also measured. The highest shear force was recorded.

#### 2.5.5. Sensory Evaluation

Sensory evaluation of SMC was conducted with 50 untrained panelists [20]. Appearance, color, odor, texture, and overall likeness scores were given on a 9-point hedonic scale, in which 1 represents ‘dislike extremely’ and 9 denotes ‘like extremely’. SMC treated without and with ME (400 mg/L) and HPP (400 MPa) at the initial (day 0) and the sample stored for 12 days with TVC lower than 6 log CFU/g were used for sensory evaluation. Prior to serving, the samples were steamed for 2 min. The panelists were asked to rinse their mouths between samples using drinking water. Sensory evaluation was approved by the ethical committee at Prince of Songkla University (ethical number: PSU-HREC-2023-007-1-1).

### 2.6. Next-Generation Sequencing (NGS)

The SMC (1 g) sample was carefully chopped using an aseptic technique to prevent contamination. Subsequently, the samples were mixed with 3 mL of a preservative agent and stored at 4 °C for analysis. The samples selected for NGS were designated as Fresh, CON, and M4P4. A fresh sample was taken to compare with CON stored for 3 days and M4P4 stored for 12 days, in which TVC did not exceed 10^6^ CFU/g. All SMC samples were processed and analyzed with the ZymoBioMICs^®^ Service (Zymo Research, Irvine, CA, USA), following the method tailored by Chayanupatkul, et al. [21]. 

### 2.7. Statistical Analysis

A completely randomized design (CRD) was used throughout the study. All experiments and analyses were carried out in triplicate. An analysis of variance (ANOVA) was conducted, and the comparison of means was performed using Tukey’s test through SPSS version 28 (SPSS Inc., Chicago, IL, USA). A level of *p* < 0.05 was used to determine statistical significance.

## 3. Results and Discussion

### 3.1. Time-Kill Profile of ME against Pseudomonas aeruginosa and Shewanella *spp.*

A time-kill assay was performed to assess the effect of ME at 400 mg/L on the counts of *P. aeruginosa* (PA-ME) and *Shewanella* spp. (SWS-ME) during 24 h in comparison with those of *P. aeruginosa* and *Shewanella* spp. without ME treatment (PA-CON and SWS-CON) (Figure 1a). At the initial time (0 h), the counts of both species were around 10^7^ CFU/mL. As the time increased, a continuous upsurge in count was observed for both PA-CON and SWS-CON. However, the counts of PA-ME and SWS-ME were continuously decreased. At the end of the storage (24 h), the counts of PA-ME and SWS-ME were 2.10 and 2.34 log CFU/mL, respectively, in which a reduction of 2.67 and 2.58 log CFU/mL was attained, respectively. PA-ME exhibited a slightly greater reduction than SWS-ME. Different bacteria plausibly had varying susceptibilities to antimicrobial agents in ME, especially during cell division. This was governed by differences in genetic material and cytoplasmic transfer [22]. The counts of PA-CON and SWS-CON reached 7.85 and 8.97 log CFU/mL, respectively, after 24 h. Different time-kill profiles for PA-ME and SWS-ME could potentially be attributed to the varying modes of action toward cell lysis by ME. Different cellular composition, cell permeability, and phase differentiation (e.g., exponential/log phase, stationary/lag phase) could determine the sensitivity of cells toward the antimicrobial agents [23]. Olatunde, et al. [24] used ethanolic coconut husk extract to inactivate *Escherichia coli*, *Staphylococcus aureus*, *Listeria monocytogenes*, and *P. aeruginosa* and found that gram-negative bacteria were more sensitive than gram-positive bacteria. This was due to the thick peptidoglycan layers of the former and different growth phases. The log phase was more sensitive toward antimicrobial agents than the lag phase. Therefore, ME had the potential to inactivate spoilage bacteria, which could be used as an antimicrobial agent for seafood products.

### 3.2. Inactivation of Planktonic Cells by HPP

Inactivation of *P. aeruginosa* and *Shewanella* spp. by HPP at 400 MPa (PA-HPP and SWS-HPP) was monitored at different HPP times (0, 1, 3, 5, and 7 min), as shown in Figure 1b. At the initial stage (0 min), the bacterial count was approximately 7 log CFU/mL. After 1 and 3 min of HPP, the count of PA-HPP and SWS-HPP was in the range of 3.67–3.87 log CFU/mL and 3.26–3.43 log CFU/mL, respectively. The counts of PA-HPP and SWS-HPP were lower than those of the initial sample (0 min). No differences in counts between those two bacteria were observed for all HPP levels used (*p* > 0.05). However, lower cell survival was noted for PA-HPP and SWS-HPP after 5 and 7 min of HPP treatment, compared to those of PA-HPP and SWS-HPP treated for 1 and 3 min (*p* < 0.05). No difference between counts of PA-HPP and SWS-HPP at 5 and 7 min was found (*p* > 0.05). The cell counts for PA-HPP and SWS-HPP after HPP for 5 min were 1.88 and 2.09 log CFU/mL, respectively, whereas the counts of those treated with HPP for 7 min were 1.75 and 2.02 log CFU/mL, respectively. The variation in viable cell counts after HPP treatment is dependent on the simultaneous effects of exposure time, temperature, pressure, and holding time, as well as bacterial cell membrane conformation, altered biochemical reactions, genetic material modification, and cell membrane permeability [9]. Microbial cell wall architecture is another major factor causing tolerance toward HPP treatment of bacteria. Bacterial cell walls are made up of peptidoglycans consisting of N-acetylglucosamine and N-acetylmuramic acid and three amino acids, i.e., D-glutamic acid, D-alanine, and meso-diaminopimelic acid, that significantly affect cell survivability [25]. Similar results were also found for *P. aeruginosa, S. aureus,* and *L. monocytogenes* when treated with HPP at 300, 400, and 500 MPa for 5, 6, and 10 min, respectively [26,27,28]. 

Overall, HPP treatment time had the potential to influence the inactivation of cells. Therefore, HPP at 400 MPa for 5 min was selected for microbial inactivation. 

### 3.3. Quality Changes of Squid Mantle Cut Treated with ME and HPP during Refrigerated Storage

#### 3.3.1. Microbiological Changes

Total viable count (TVC), psychrophilic bacteria count (PBC), *Pseudomonas* spp. count (PDC), H_2_S-producing bacteria count (HSPBC), and *Enterobacteriaceae* count (EBC) of SMC treated with ME at 0, 200, and 400 mg/L in conjunction with HPP at 200 and 400 MPa during 15 days of refrigerated storage are shown in Figure 2a–e. At day 0, the TVC of CON was 2.98 log CFU/g, whereas all samples treated with HPP at 400 MPa had no count detected (*p* > 0.05). After 3 days of storage, the TVC of all samples treated with HPP increased, indicating that the microbial cell damage induced by HPP was reversible. The HPP-treated cells were able to repair themselves, leading to increased growth with prolonged storage. Lower TVC was observed in M2P4 and M4P4 samples, compared to that of CON (*p* < 0.05). Pathanasriwong, et al. [29] reported that variation in TVC was contingent upon the composition of the raw materials, storage parameters, duration of treatment, microbial acclimatization, utilization of additives, and enzymatic activity. Teixeira, et al. [30] documented that extensive variations in TVC may impact preservation and shelf-life. Implementation of hurdles can enhance the potential for extending the shelf-life of seafood [31]. At day 6, TVC of CON, M0P2, and M0P4 exceeded the acceptable limit, indicating that HPP treatment alone could not retard the microbial proliferation to a high extent. Additionally, M4P4 had the highest reduction of TVC (6.26 log CFU/g) compared to the rest of the samples after 12 days of storage (*p* < 0.05). The lower bacterial count might be due to the combined effect of ME (400 mg/L) and HPP (400 MPa). The acceptable limit of TVC in seafood is 6 log CFU/g, but the better-quality product is considered within the limit of 5 log CFU/g [14]. From our previous study, LC-MS data for ME identified major phenolic compounds that plausibly played a significant role as antimicrobial compounds [14]. The ME was dominated by phenolic compounds, including flavonoids, tannins, and phenolic acids, known for their significant antimicrobial properties. Flavonoids can form complexes with bacterial extracellular proteins and cell walls, while tannins impede microbial adhesion and inactivate enzymes and cell envelope proteins [32]. Tannins inhibit extracellular microbial enzymes by depriving the substrates required for microbial proliferation or directly affecting microbial metabolism by inhibiting oxidative phosphorylation related to the inhibition of ATP synthesis, ultimately leading to cell death [33]. On the other hand, plant extract in combination with HPP caused microbial cell damage and inactivation to a higher degree. HPP, along with other hurdles, such as plant extracts, CO_2_, and nisin (10 µg/g), can retard bacterial growth more effectively than HPP alone [34,35]. 

The psychrophilic bacteria count (PBC) for samples treated with ME at 200 and 400 mg/L and HPP (200 and 400 MPa) was augmented, but at a lower increasing rate than the CON (*p* < 0.05) after day 3. However, at day 6, the increase of PBC was varied among all samples (*p* < 0.05), in which the CON exceeded 6 log CFU/g by reaching 6.58 log CFU/g. Both ME and HPP were able to retard the growth during storage at refrigerated temperatures. Due to the wide range of temperature tolerance (4 to 25 °C), psychrophilic bacteria can easily survive and proliferate [36].

A lower PDC was found in the presence of ME along with HPP treatment (*p* < 0.05). CON had the highest count on day 6 (6.18 log CFU/g). PDCs for M2P2, M2P4, and M4P2 were 6.48, 6.79, and 6.63 log CFU/g at day 12, respectively. Although the count was increased for all the treated samples (*p* < 0.05), the increasing rate was less than that of CON (*p* < 0.05). Chill-stored fish and cephalopods such as cuttlefish and shortfin squid underwent spoilage as induced by *Pseudomonas* spp. [37,38,39,40]. Pletzer, et al. [41] observed that *Pseudomonas* spp. can grow slowly at 3 °C, and approximately 46% of fish spoilage was caused by these microorganisms. In this study, the reduced rate of growth of *Pseudomonas* spp. was possibly due to the HPP treatment conducted at a low processing temperature (4 °C), which could alter cell wall lipid composition, making them more sensitive [42]. ME is a good source of hydroxycinnamic acid derivatives such as *p*-coumaric, ferulic, and caffeic acids. These compounds act as gram-negative bacterial cell-blocking agents [43]. Therefore, the application of natural extract (ME) and pressurization of SMC exhibited a combined effect on reducing the proliferation of *Pseudomonas* spp. during refrigerated storage.

The H_2_S-producing bacteria (HSPB) are responsible for H_2_S gas production. They can reduce trimethylamine oxide (TMAO) to trimethylamine (TMA), causing off-odors [44]. At day 0, CON had an HSPBC of 2.22 log CFU/g, whereas the counts of M2P4 and M4P4 were not detected. At the end of the storage (15 days), M4P4 showed the lowest count compared to others (*p* < 0.05). In the HSPB group, the dominant microflora found during seafood spoilage include *Shewanella putrificiences* and *S. baltica*. Most of them are gram-negative and can grow at refrigerated temperatures (4–10 °C). Additionally, *Photobacterium* and *Shewanella* produce H_2_S by the breakdown of sulfur-containing amino acids (cysteine, methionine-methyl-mercaptan, dimethyl-disulfide) causing strong off-odors and spoilage in seafood [45]. However, the HPP treatment considerably reduced the growth of HSPB at specified pressures (300 and 500 MPa, 5 min) in oysters, as observed by Cruz-Romero, et al. [46]. Similarly, Diachkova, et al. [47] isolated microorganisms from cephalopods, causing the formation of sulfur-producing amino acids. The HPP-treated mackerel fillets had less HSPBC than the control because the initial load was reduced and less production of bacterial metabolites was achieved [48]. As storage progressed, several sulfur-containing volatile compounds (dimethyl and trimethyl sulfide), organic acids, and ammonia were developed, leading to a sour taste in cuttlefish stored at 4 °C. These compounds increased the pH, thus supporting bacterial growth and reducing the firmness of cuttlefish [47]. 

For *Enterobacteriaceae*, all samples had EBC ranging from 2.01 to 7.23 log CFU/g throughout the storage (*p* < 0.05). At the beginning, all treated samples had no *Enterobacteriaceae* detected (ND), except for the CON sample (3.34 log CFU/g). This indicated that the initial microbial load was reduced by the combined effect of the hurdles used. As storage time progressed to day 6, all three samples, including CON, M0P2, and M0P4, had EBC greater than 6 log CFU/g. In contrast, the other four samples combined with ME and HPP (M2P2, M2P4, M4P2, and M4P4) had reduced bacterial proliferation (*p* < 0.05). Among all samples, M4P4 showed the highest reduction of microbial growth. ME at a high concentration (400 mg/L) and HPP (400 MPa) have a combined impact on the retardation of bacterial cell growth. The result was similar to psychrophilic bacteria (PBC) and *Pseudomonas* spp. growth, where M4P4 had the least bacterial count compared to the rest of the samples (*p* < 0.05). *Enterobacteriaceae* consist of *Shewanella* spp. and *Moraxella* spp. commonly found in seafood and majorly cause spoilage and off-odors [49]. Zhuang, et al. [50] revealed that *S. putrefaciens* hydrolyzed collagen and degraded both thick and thin filaments of myofibrillar proteins in grass carp. HPP in the range of 200–300 MPa can destroy microbial cell wall by altering enzymatic activity and mitochondrial structure [51]. *Enterobacter aerogenes* produces acetoin from pyruvic acid to raise the pH of its growth environment. Furthermore, when amino acids are decarboxylated, an increase in pH occurs from the resulting amines [45]. 

#### 3.3.2. Total Volatile Base (TVB) and Trimethylamine (TMA) Contents

TVB and TMA contents of SMC treated with ME at various concentrations and subjected to HPP at different levels are depicted in Figure 3a,b. TVB content is one of the spoilage indexes, where ammonia, dimethylamine, and trimethylamine are produced due to protein decomposition. Bekhit, et al. [52] reported that the maximum allowable TVB value was between 25 and 35 mg N/100 g sample for seafood. In squid meat, TVB content rose as spoilage increased over time [7]. At day 0, the TVB contents of all samples were in the range of 2.11–2.35 mg N/100 g. No differences in TVB contents were observed among all samples (*p* > 0.05). After 6 days of storage, CON had a higher TVB content (25.87 mg N/100 g sample) than all treated samples (*p* < 0.05). After day 12, the TVB contents of M2P2, M4P2, M2P4, and M4P4 samples drastically increased, and the values of 20.00, 21.23, 18.00, and 15.21 mg N/100 g samples were obtained, respectively. However, these values were lower than those of CON throughout the storage (*p* < 0.05). Nevertheless, the TVB content of M4P4 was the lowest, compared to that of CON and other treated samples (*p* < 0.05). Thus, samples treated with 400 mg/L, followed by a 400 MPa HPP treatment, provided the hurdles and retarded microbial growth and protein decomposition. Tantasuttikul, et al. [53] and Palamae, Mittal, Buatong, Zhang, Hong, and Benjakul [15] reported that TVB content was reduced by 1.2 times in blood clams, seabass fillets, mackerel fillets, and green mussels when HPP at 300–500 MPa was applied for 3 min. The HPP-treated shrimp showed lower TVB content than thermally treated samples. HPP-treated seabass fillets (300 MPa) showed lower TVB formation than the control and remained lower than the limit after 18 days of iced storage [30]. In addition, pressure level, processing time, and the initial condition of the muscle also determined an increase in TVB content [7,54,55]. However, the treatment of SMC using ME at 400 mg/L in combination with HPP at 400 MPa lowered microbial activity, as witnessed by the lower TBV content.

The TMA content of SMC as affected by the treatment of ME and HPP was monitored throughout the refrigerated storage period of 15 days. Trimethylamine and dimethylamine are generated by microorganisms and autolytic enzymes and are considered the spoilage indicators of fishery products [56]. At day 0, TMA content for all treated samples was found within the range of 2.11–2.31 mg N/100 g sample, whereas the CON had TMA content of 2.98 mg N/100 g sample. As the storage time augmented, a continuous increase in TMA content was observed in all treated samples (*p* < 0.05). The variation in TMA content was evident based on the treatments. The high concentration of ME and high pressure at 400 MPa (M4P4) resulted in a lower level of TMA in treated SMC till the end of the storage (7.18 mg N/100 g sample). This indicated that HPP at high levels could retard TMA formation to a greater extent. Thus, the inhibitory effect of ME against microbial growth was augmented at high concentrations. Vaz-Pires, Seixas, Mota, Lapa-Guimarães, Pickova, Lindo, and Silva [39] documented that cuttlefish samples kept on ice for 15 days still had TMA below the acceptable limit. TMAO is reduced to TMA due to the activity of microbial intervention [44]. Some gram-negative bacteria are inherent in seafood, such as *S. putrificiences*, *P. phosphoreum*, *Aeromonas* spp., *Enterobacteriaceae,* and *Vibrio* spp. They are actively involved in decarboxylation to produce biogenic amines and nucleotide metabolites. In addition, these microorganisms could induce the formation of TMA. Parlapani, Michailidou, Anagnostopoulos, Sakellariou, Pasentsis, Psomopoulos, Argiriou, Haroutounian, and Boziaris [38] found that cuttlefish (*Sepia officinalis*) was rejected with a TMA content of 28.26 mg N/100 g sample after 15 days of iced storage. The free amino acid contents increased gradually in squid, cuttlefish, and octopus with increasing storage time, and this might be due to the hydrolysis of mantle and connective tissue proteins [57]. These degradation products could be utilized by spoilage bacteria with ease, in which TMA could be formed. ME (400 mg/L) and HPP (400 MPa) could retard the formation of TVB and TMA in SMC. 

#### 3.3.3. pH

pH is an indicator to determine the freshness of squid [58]. The pH of treated samples combined with ME (0, 200, and 400 mg/L) and HPP (200 and 400 MPa) during refrigerated storage is depicted in Figure 3c. Overall, the pH of all samples increased with rising storage time (*p* < 0.05). At day 0, a similar pH value was found for all SMC samples (*p* > 0.05). After 3 days of storage, a lower pH value was observed for M4P4, compared to those of other SMC samples (*p* < 0.05). The highest increasing rate of pH occurred in CON (*p* < 0.05). The sample treated with 400 mg/L ME and 400 MPa HPP had a stable pH. This might be related to the lower growth of microorganisms in this sample since ME and HPP at high levels could inactivate spoilage bacteria. Olsson, et al. [59] found that the lower pH changes supported less bacterial proliferation in halibut muscle during cold storage. After 15 days of storage, M4P4 had the lowest pH value (7.45) (*p* < 0.05). HPP not only assisted the buffering capacity to remain stable but also caused minimum changes in raw material composition [60]. After HPP treatment, the microbial membrane permeability increased with the higher intensity of the HPP level applied (500 MPa, 5 min), leading to bacterial inactivation [17]. The intracellular pH of pressure-stressed microbial cells decreased by 0.2 units, and ATPase synthesis activity was markedly reduced [61]. Therefore, active transport of protons out of the cell cannot take place, and the bacteria die due to acidification [62]. 

#### 3.3.4. PV and TBARS Value

PV is used to evaluate the primary oxidation products throughout the storage period (Figure 3d). At day 0, a higher PV was observed for the CON (*p* < 0.05) compared to those of other samples. After 3 days of storage, a slight increase in PV was attained for all samples. However, after 9 days, a drastic augmentation in PV was found for all SMC samples, especially in the CON, M0P2, and M0P4 samples. Among all samples, the M4P4 showed lower PV (8.01 mg cumene hydroperoxide/kg sample) than the CON (21.25 mg cumene hydroperoxide/kg sample) at the end of the storage (*p* < 0.05). However, all values were below the acceptable limit. Howgate [56] found that the PV of octopus without any treatment increased continuously during iced storage. For the present study, primary lipid oxidation products in SMC might be reduced because ME had antioxidant activity. Linarin, kaempferol 3-O-rutinoside, quercetin, and caffeic acid in ME more likely retarded the free radical generation in all treated samples [14]. Among these phenolic compounds, linarin is a flavonoid that serves as an antioxidant agent. Its structure carries two OH- groups, which could terminate the propagation stage [43]. In addition, HPP treatment probably inactivated oxidative enzymes in SMC. This indicated the combined effect of ME and HPP on delaying lipid oxidation. SMC contained polyunsaturated fatty acids (PUFAs) that were prone to oxidation, resulting in the formation of a mixture of aldehydes, epoxides, and ketones associated with the development of rancid flavor/odor [63]. PV of HPP-treated clams (300 and 400 MPa, 5 min) was changed slowly during iced storage, as reported by Vaz-Pires, Seixas, Mota, Lapa-Guimarães, Pickova, Lindo, and Silva [39]. HPP above 500 MPa can cause tissue or muscle rupture, exposing the internal components to oxygen. This can induce lipid oxidation, which leads to the production of volatile compounds [23].

Immediately after processing (day 0), no marked differences were observed in TBARS values for both the treated and control samples (Figure 3e). The TBARS value of the CON was higher than that of other samples treated with HPP at 200 and 400 MPa on days 3 and 6 (*p* < 0.05). The TBARS value of all samples increased up to day 9, while a decreasing rate was observed at days 12 and 15, respectively. Throughout the storage, the minimum amount of TBARS formation was achieved for M2P4 and M4P4 (0.61 and 0.42 mg MDA equivalent/kg sample). This gradual increase in the TBARS value indicated the formation of secondary oxidation products. This might be associated with the hindered formation of primary oxidation products by the combined effect of ME and HPP. The bioactive compounds present in the ME mitigate the lipid oxidation or deterioration via autooxidation, while HPP could lower the enzyme-inducing lipid oxidation or inactivate lipase, phospholipase, etc. 

#### 3.3.5. Weight and Cooking Losses

Generally, weight loss and cooking loss increased continuously as the storage time upsurged (*p* < 0.05) (Table 1). After 3 days, the weight loss of all samples was in the range of 2.45–2.57%. The samples without ME treatment had similar (*p* > 0.05) weight loss to the CON, regardless of the HPP level used. However, both samples treated with 200 and 400 mg/L and HPP at 200 MPa (M2P2 and M4P2) had lower weight loss than those of samples treated with 200 and 400 mg/L and HPP at 400 MPa (M2P4 and M4P4) (*p* < 0.05). After 6 days, lower weight loss was observed for all samples treated with HPP at 400 MPa, especially when a higher concentration of ME was used (M4P4). During 6–15 days of storage, the lowest increasing rate of weight loss was found in the M4P4 sample (*p* < 0.05). At the end of the storage (15 days), M4P4 had a weight loss of 7.18%. The weight loss was most likely caused by the lower water-holding capacity and protein denaturation and degradation caused by autolysis and microbial enzyme activity [64]. Bouletis, et al. [65] documented that cuttlefish treated with HPP (400 MPa, 3 min) and stored in ice had a minimum reduction in weight loss (5%) compared to the control (11%). Weight loss also depends on the species, initial quality, storage conditions, and duration of storage [66]. 

At day 0, the cooking loss of all samples was in the range of 7.09–7.78%. Lower weight loss was observed for the samples treated with ME at 400 mg/L (*p* < 0.05) compared to those of samples treated without and with 200 mg/L ME, irrespective of the level of HPP used. As the storage time upsurged, a high augmenting rate of cooking loss was obtained for the CON sample. The lowest increasing rate of cooking loss was found in the M4P4 sample (*p* < 0.05). After 15 days of storage, M4P4 had a cooking loss of 19.17%, suggesting the highest water-holding capacity (WHC) of SMC. The changes in pH also determined the cooking loss of muscle [66]. Due to the lower pH, the firmness of SMC proteins (myofibrillar proteins) remains intact, which could lead to less cooking loss. A similar result was reported by Temdee, Singh, and Benjakul [18] in mantis shrimp during 10 days of iced storage. Cooking loss in beef was reduced by 12% after HPP treatment at 300 and 400 MPa, compared to that found in the samples without HPP.

#### 3.3.6. Textural Property

The firmness and hardness of raw SMC decreased continuously during the entire storage (*p* < 0.05) (Figure 4a,b), which was in line with the increases in weight loss and cooking loss (Table 1). In general, the textural property of SMC could be impacted by various factors (pH, degradation of myofibrils caused by indigenous or microbial proteases, lipid and protein oxidation) [67]. At day 0, the firmness of all samples was insignificant, ranging from 7658.84 g to 7005.30 g. However, firmness decreased with increasing storage time. The CON (1768.10 g) showed a lower firmness than that of the M4P4 sample (2868.00 g) on the final day (15 days) (*p* < 0.05). Among all treatments, toughness showed a similar trend to firmness (*p* < 0.05). This indicated the combined effect of ME and HPP in maintaining the texture. Both treatments could lower microbial growth and inactivate proteases from microorganisms or indigenous proteases in SMC. Zhang, et al. [68] reported lower textural alteration when the squid mantle was subjected to HPP at 200 and 400 MPa for 10 min. Mantle and muscle tissue collagen and myofibrillar proteins were not degraded when those pressure levels were applied. Processing temperature and holding time accounted for the changes in texture [69,70].

#### 3.3.7. Sensorial Property

The likeness scores of selected samples at days 0 and 12 of storage are shown in Table 2. At day 0, no differences in likeness scores were found between CON and M4P4 for all attributes tested (*p* < 0.05). At day 12, the CON had TVC exceeding the limit and was not used for evaluation, whereas M4P4 still remained acceptable TVC. After 12 days of storage, M4P4 had an overall likeness score of 6.45. However, the decrease in likeness was in line with the increased TBV content and TBARS value. ME and HPP showed a combined effect on the reduction of the formation of volatile compounds resulting from nitrogenous and non-nitrogenous compound degradation. Also, bacterial cell growth as well as oxidative deterioration decreased. Furthermore, the sensory attributes of M4P4 indicated that HPP at 400 MPa resulted in less alteration of color, odor, and texture. This effect might be due to the inactivation of enzymatic activity in SMC, leading to lower autolytic activity and bacterial growth [71]. In addition, sensory properties are correlated with water-holding capacity, where M4P4 showed the highest sensory properties. Moreover, at 400 MPa, reversible protein denaturation might occur, and hydrogen bonds were less affected in the myofibrillar protein, which is responsible for protein hydration. This led to preserving the muscle texture and flavor [72]. Therefore, it could be stated that the combined process between ME at 400 mg/L and HPP at 400 MPa could preserve the sensory attributes of SMC during storage.

### 3.4. Microbial Community in SMC and the Selected ME and HPP-Treated SMC

The bacterial diversity in fresh SMC (day 0), CON (subjected to 0.1 MPa and kept for 3 days), and M4P4 (treated with ME at 400 mg/L, followed by HPP at 400 MPa and stored at 4 ºC for 12 days) is depicted in Figure 5a,b. 16S rRNA gene sequencing was used to identify the microbial assemblage responsible for SMC spoilage. Microbial taxonomy, including family, genus, and species levels, was reported. In the fresh sample, the predominant families consisted of *Listeriaceae* (53.72%), *Vibrionaceae* (42.61%), *Psychrmonadaceae* (2.68%) and *Moraxellaceae* (1.02%). Two species were *Brochothrix campestrix* and *Photobacterium kishitanii*, which are related to the family *Listeriaceae*. The prevalent species were *Brochothrix campestris* (53.26%), *Photobacterium kishitanii* (25.29%), *Photobacterium iliopiscarium* (10.24%), *Aliivibrio salmonicidaz* (3.48%), *Psychromonas arctica* (2.68%), *Photobacterium piscicola* (1.79%), and other species (3.22%) were found in the fresh sample. However, at day 3 of storage of the CON sample, the abundance of family *Vibrionaceae* (84.31%) became dominant instead of *Listeriaceae*. *Listeriaceae* (5.98%), *Fusobacteriaceae* (5.02%), *Moraxellaceae* (0.99%) and other families (0.73%), were observed. It is noteworthy that *Enterococcaceae* (2.97%) were found in a CON sample. For CON after 3 days, *Photobacterium kishitani* (33.3%) and *Photobacterium piscicola* (22.5%), related to the family *Vibrionaceae*, were obtained. *Photobacterium iliopiscarium* (8.74%), *Aliivibrio-Vibrio* sp. 64772 (7.36%), *Brochothrix campestris* (5.98%), *Psychrilyobacter atlanticus* (5.02%), *Aliivibrio fischeri-sifiae* (4.49%), *Photobacterium frigidiphilum* (2.77%), *Vagococcus fessus* (2.52%), *Aliivibrio fischeri* (1.08%) and other species (6.19%) were also detected. After day 12 of the sample treated with 400 mg/L ME (M4P4), the family *Listeriaceae* (96.85%) became dominant. *Carnobacteriaceae* (1.97%) and other families (1.18%) were found at low proportions. *Brochothrix campestris* (96.84%), *Psychrilyobacter atlanticus* (0.01%), and other species (3.08%) were identified in the M4P4 sample. Among all samples, relative abundance revealed that *Listeriaceae* was the predominant family, and a 48% reduction in abundance was found in the CON after day 3 of storage. M4P4 showed the highest increase in abundance of *Listeriaceae* when compared with fresh and CON samples (43.58% and 90.87%, respectively). The increase in *Listeriaceae* (96.84%) in the M4P4 sample might be due to the application of ME (400 mg/L) and HPP levels at 400 MPa, which inhibited other microorganisms. Phytochemicals have a unique nature to disrupt the cell walls of gram-negative and gram-positive bacteria, along with HPP at 400 MPa. Bacterial cell leakage triggered their deaths. As the storage time augmented, an increasing rate of bacterial growth was detected. It was also reported that two species of *Listeriaceae* caused seafood spoilage [64]. *Brochothrix campestris* and *Brochothrix thermosphacta* were the main bacteria that caused squid spoilage at low storage temperatures [55]. They are gram-positive rods and facultative anaerobes, which produce off-odors and off-flavors by reducing TMAO into TMA [73]. In the present study, ME might have less effectiveness in inactivating *Brochothrix campestris* during storage. HPP played a major role in damaging the cells; however, recovery could take place during storage [14]. The squid samples were dominated by species that caused spoilage during storage. *Photobacterium*, *Brochothrix*, and *Carnobacterium* were the potential dominant genera in the spoilage of cuttlefish [73,74]. At HPP levels of 200 and 400 MPa for 5 and 10 min, changes in the metabarcoding analysis showed different species causing the spoilage in different molluscs [55]. Moreover, storage temperature and thermal shock enhanced cell damage [75]. *Moraxellaceae* were completely reduced in M4P4, while they were found in the fresh sample and CON sample stored for 3 days, ranging from 0.99% to 1.02%. The reduction rate was also similar to *Listeriaceae*. *Moraxillaceae* were identified as being responsible for food and seafood quality deterioration [42,76].

Among all samples, three main bacterial species (*Brochothrix campestris*, *Photobacterium kishitanii*, and *Photobacterium piscicola*) were detected at high relative abundances, and the other five species were found in traces (*Photobacterium iliopiscarium*, *Aliivibrio fischeri*, *Psychrilyobacter atlanticus*, *Vagococcus fessus*, and *Psychromonas arctica*) during the entire storage period. *Photobacterium kishitanii* and *Photobacterium piscicola* are proteobacteriacae responsible for the degradation of non-nitrogenous compounds at low temperatures (0–10 °C), leading to volatile compound formation [44,64]. 

Mint extract and HPP showed effectiveness in reducing microbial spoilage by inhibiting specific bacteria during extended storage, as shown by next-generation sequencing. This sequencing validated the bacterial diversity responsible for SMC spoilage when mint extract and HPP were applied.

## 4. Conclusions

ME and HPP applied for SMC treatment effectively retarded the growth of mesophilic and psychrophilic bacteria counts, *Pseudomonas* spp. H_2_S-producing bacteria, and *Enterobacteriaceae* counts. Spoilage index, lipid oxidation, and softening of texture were also impeded, particularly when ME and HPP at high levels were employed. The next-generation sequencing indicated that the predominant family was *Listeriaceae*. *Brochothrix campestris* was responsible for spoilage in M4P4 kept for 12 days at 4 °C. Overall, ME at 400 mg/L and HPP at 400 MPa were the combined hurdles for the shelf-life extension of SMC. Treatment of SMC in combination with ME and HPP also improved sensory attributes.

## Figures and Tables

**Figure 1 foods-13-01264-f001:**
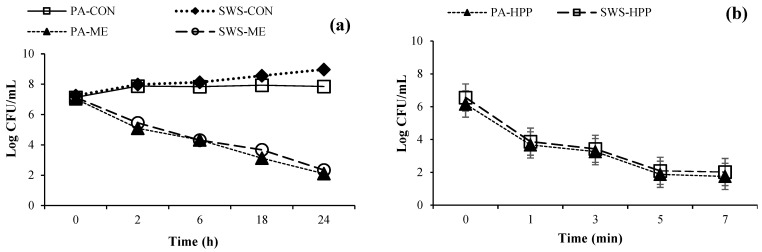
Time-kill profile as function of time of mint extract at a concentration of 400 mg/L against *Pseudomonas aeruginosa* (PA) and *Shewanella* spp. (SWS) (**a**) and planktonic cell inactivation as function of time of high-pressure processing (HPP) at 400 MPa against PA and SWS (**b**). Bars represent the standard deviation (*n* = 3). PA-CON and PA-ME denote the PA treated without and with mint extract (ME), respectively. SWS-CON and SWS-ME denote the SWS treated without and with ME, respectively.

**Figure 2 foods-13-01264-f002:**
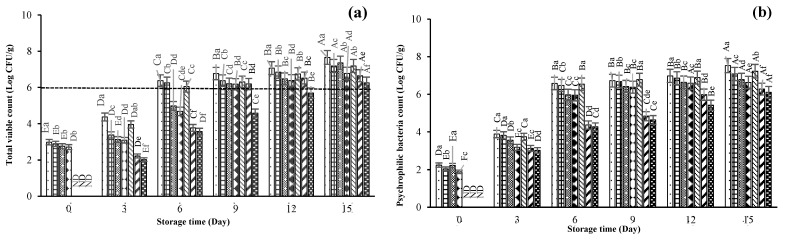
Changes in total viable count (TVC) (**a**); psychrophilic bacteria count (PBC) (**b**); *Pseudomonas* spp. count (PDC) (**c**); H_2_S-producing bacteria count (HSPBC) (**d**); and *Enterobacteriaceae* count (EBC) (**e**) of squid mantle cut treated with mint extract at different concentrations, followed by high-pressure processing at various levels during refrigerated storage for 15 days. Bars represent the standard deviation (*n* = 3). Different uppercase letters on the bars within the same treatment indicate significant differences (*p* < 0.05). Different lowercase letters on the bars within the same storage time indicate significant differences (*p* < 0.05). ND: Not detected. The dashed line represents the total viable count (TVC) limit of 6 log CFU/g. CON denotes the squid mantle cut (SMC) without mint extract (ME) and high pressure processing (HPP); M0P2, M2P2, and M4P2 denote the SMC treated with ME at concentrations of 0, 200, and 400 mg/L, respectively, followed by HPP at 200 MPa; M0P4, M2P4, and M4P4 denote the SMC treated with ME at concentrations of 0, 200, and 400 mg/L, respectively, followed by HPP at 400 MPa.

**Figure 3 foods-13-01264-f003:**
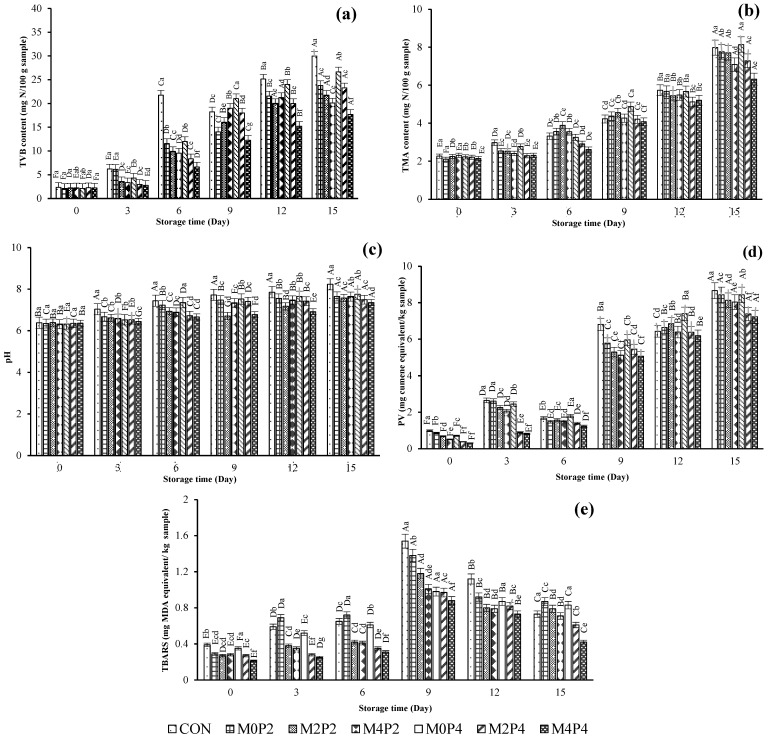
Changes in total volatile base (TVB) content (**a**); trimethylamine (TMA) content (**b**); pH (**c**); peroxide value (PV) (**d**); and thioburbituric acid reactive substances (TBARS) (**e**) of squid mantle cut treated with mint extract at different concentrations, followed by high-pressure processing at various levels during refrigerated storage for 15 days. Bars represent the standard deviation (*n* = 3). Different uppercase letters on the bars within the same treatment indicate significant differences (*p* < 0.05). Different lowercase letters on the bars within the same storage time indicate significant differences (*p* < 0.05). CON denotes the squid mantle cut (SMC) without mint extract (ME) and high pressure processing (HPP); M0P2, M2P2, and M4P2 denote the SMC treated with ME at concentrations of 0, 200, and 400 mg/L, respectively, followed by HPP at 200 MPa; M0P4, M2P4, and M4P4 denote the SMC treated with ME at concentrations of 0, 200, and 400 mg/L, respectively, followed by HPP at 400 MPa.

**Figure 4 foods-13-01264-f004:**
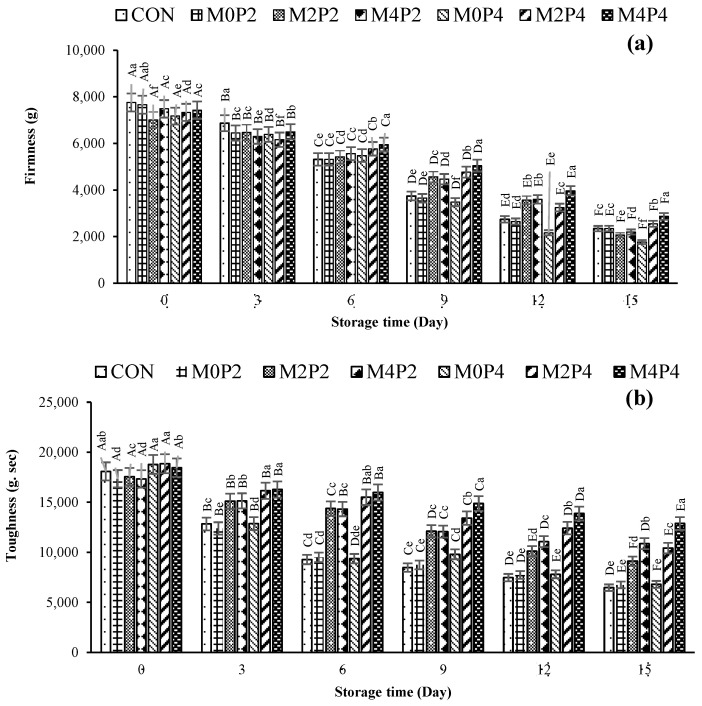
Changes in firmness (**a**) and toughness (**b**) of squid mantle cut treated with mint extract at different concentrations followed by high-pressure processing at various levels during refrigerated storage for 15 days. Bars represent the standard deviation (*n* = 3). Different uppercase letters on the bars within the same treatment indicate significant differences (*p* < 0.05). Different lowercase letters on the bars within the same storage time indicate significant differences (*p* < 0.05). CON denotes the squid mantle cut (SMC) without mint extract (ME) and high pressure processing (HPP); M0P2, M2P2, and M4P2 denote the SMC treated with ME at concentrations of 0, 200, and 400 mg/L, respectively, followed by HPP at 200 MPa; M0P4, M2P4, and M4P4 denote the SMC treated with ME at concentrations of 0, 200, and 400 mg/L, respectively, followed by HPP at 400 MPa.

**Figure 5 foods-13-01264-f005:**
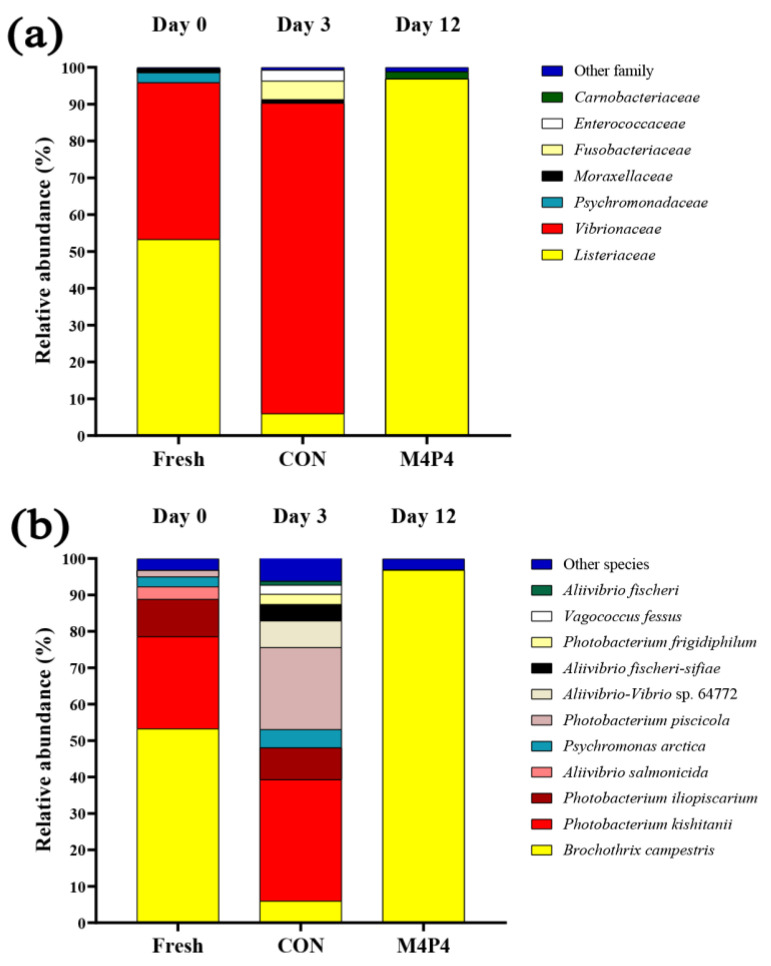
Relative abundance (%) of the taxonomy at family (**a**) and species (**b**) level of the bacteria identified in fresh squid mantle cut, the control (without any treatment) stored for 3 days, and those treated with ME at 400 mg/L and HPP at 400 MPa for 5 min and stored for 12 days. Storage was performed at 4 °C. Unassigned and low abundant species (<1%) were in the group of “others”.

**Table 1 foods-13-01264-t001:** Changes in weight loss and cooking loss of squid mantle cut treated with mint extract at different concentrations and high-pressure processing at various levels during refrigerated storage.

Storage Time (Day)	Weight Loss (%)						
	CON	M0P2	M2P2	M4P2	M0P4	M2P4	M4P4
0	---	---	---	---	---	---	---
3	2.53 ± 0.05 *^Be^	2.51 ± 0.01 ^Be^	2.45 ± 0.06 ^De^	2.48 ± 0.09 ^Ce^	2.51 ± 0.04 ^Be^	2.57 ± 0.06 ^Ae^	2.52 ± 0.02 ^Be^
6	4.76 ± 0.47 ^Bd^	4.65 ± 0.04 ^Cd^	5.32 ± 0.09 ^Ad^	5.34 ± 0.07 ^Ad^	3.98 ± 0.08 ^Dd^	3.75 ± 0.08 ^Ed^	3.03 ± 0.03 ^Fd^
9	7.86 ± 0.18 ^Bc^	7.67 ± 0.02 ^Cc^	6.81 ± 0.02 ^Dc^	7.89 ± 0.02 ^Ac^	5.92 ± 0.05 ^Ec^	5.89 ± 0.09 ^Fc^	4.35 ± 0.04 ^Gc^
12	9.96 ± 0.31 ^Ab^	8.78 ± 0.03 ^Bb^	8.22 ± 0.04 ^Db^	8.56 ± 0.06 ^Cb^	7.84 ± 0.06 ^Eb^	7.74 ± 0.07 ^Fb^	6.51 ± 0.07 ^Gb^
15	11.67 ± 0.37 ^Aa^	9.95 ± 0.02 ^Ba^	9.76 ± 0.05 ^Ca^	9.43 ± 0.02 ^Da^	9.22 ± 0.07 ^Ea^	9.04 ± 0.05 ^Fa^	7.18 ± 0.09 ^Ga^
**Storage Time (Day)**	**Cooking Loss (%)**						
	**CON**	**M0P2**	**M2P2**	**M4P2**	**M0P4**	**M2P4**	**M4P4**
0	7.71 ± 0.21 *^Bf^	7.31 ± 0.41 ^Cf^	7.35 ± 0.53 ^Cf^	7.29 ± 0.73 ^Df^	7.78 ± 0.63 ^Af^	7.22 ± 0.23 ^Ef^	7.09 ± 0.53 ^Ff^
3	9.52 ± 0.15 ^Be^	9.65 ± 0.31 ^Ae^	8.12 ± 0.45 ^Ee^	8.52 ± 0.92 ^De^	8.68 ± 0.84 ^Ce^	7.55 ± 0.85 ^Ge^	7.95 ± 0.63 ^Fe^
6	12.76 ± 0.26 ^Ad^	11.89 ± 0.21 ^Bd^	10.38 ± 0.32 ^Dd^	10.15 ± 0.71 ^Ed^	11.52 ± 0.34 ^Cd^	9.82 ± 0.24 ^Gd^	9.87 ± 0.83 ^Fd^
9	17.95 ± 0.27 ^Ac^	17.48 ± 0.24 ^Bc^	15.65 ± 0.24 ^Cc^	15.36 ± 0.63 ^Cc^	17.38 ± 0.21 ^Bc^	12.71 ± 0.17 ^Dc^	12.64 ± 0.75 ^Dc^
12	23.67 ± 0.21 ^Bb^	23.15 ± 0.23 ^Cab^	19.56 ± 0.12 ^Db^	19.13 ± 0.73 ^Eb^	24.18 ± 0.63 ^Ab^	17.44 ± 0.32 ^Fa^	17.34 ± 0.43 ^Gb^
15	30.78 ± 0.28 ^Aa^	25.45 ± 0.31 ^Ca^	23.39 ± 0.56 ^Da^	23.21 ± 0.12 ^Ea^	28.45 ± 0.63 ^Ba^	21.73 ± 0.52 ^Ea^	19.71 ± 0.73 ^Ga^

* The value represents the mean ± SD. Different lowercase superscripts within the same column of weight loss or cooking loss indicate significant differences (*p* < 0.05). Different uppercase superscripts within the same row indicate significant differences (*p* < 0.05). CON denotes the squid mantle cut (SMC) without mint extract (ME) and high pressure processing (HPP); M0P2, M2P2, and M4P2 denote the SMC treated with ME at concentrations of 0, 200, and 400 mg/L, respectively, followed by HPP at 200 MPa; M0P4, M2P4, and M4P4 denote the SMC treated with ME at concentrations of 0, 200, and 400 mg/L, respectively, followed by HPP at 400 MPa.

**Table 2 foods-13-01264-t002:** Changes in likeness score of selected squid mantle cut treated without and with mint extract at 400 mg/L and high-pressure processing at 400 MPa stored at days 0 and 12.

Storage Time (Day)	Samples	Appearance	Color	Odor	Texture	Overall Acceptance
0	CON	8.07 ± 0.03 *^a^	8.78 ± 0.13 ^a^	8.77 ± 0.09 ^a^	8.75 ± 0.10 ^a^	8.61 ± 0.19 ^a^
	M4P4	8.52 ± 0.02 ^a^	8.91 ± 0.11 ^a^	8.79 ± 0.06 ^a^	8.89 ± 0.12 ^a^	8.67 ± 0.21 ^a^
12	M4P4	7.12 ± 0.21 ^a^	7.73 ± 0.15 ^a^	6.03 ± 0.01 ^a^	6.83 ± 0.26 ^a^	6.45 ± 0.03 ^a^

* The value represents the mean ± SD. Different lowercase superscripts within the same column indicate significant differences (*p* < 0.05). CON denoted the SMC without ME treatment and HPP; M4P4 denoted the SMC treated with ME at a concentration of 400 mg/L, followed by HPP at 400 MPa.

## Data Availability

The original contributions presented in the study are included in the article, further inquiries can be directed to the corresponding author.

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
