# Peer review of "Quality of Refrigerated Squid Mantle Cut Treated with Mint Extract Subjected to High-Pressure Processing"

_foods, 2024, doi:10.3390/foods13081264_

Round 1

Reviewer 1 Report

Comments and Suggestions for Authors

1- Please, consider an updated title

2- Please, update the section of introduction with recent information from recent References (2022-2024), to make your study more interesting and more significant to the reader.

for ex. Coupling ozone with microbubbles (OMB) water for food disinfection: Effects on microbiological safety, physicochemical quality, and reducing pink discoloration of jumbo squid (Dosidicus gigas) - ScienceDirect

Evaluating the effects of nanoparticles combined ultrasonic-microwave thawing on water holding capacity, oxidation, and protein conformation in jumbo squid (Dosidicus gigas) mantles - ScienceDirect

Animals | Free Full-Text | Innovative Seafood Preservation Technologies: Recent Developments (mdpi.com)

3- In the section 2.2. and 2.4 please try to edit these steps referring to their references sources in order to be clearer and more scientific. Please write these steps from the ref.

4- Please, provide a better figures of 2,3 and 4 of higher resolution, because the current figures  is not clear

Comments on the Quality of English Language

Dear ,

 this research needs a simple linguistic review, and it is written in good language, but it is better for it to be reviewed by a native English speaker.

Kind regards

Samir

Author Response

Response to reviewer

****Thank you so much for the comment. All queries have been responded.

1- Please, consider an updated title

***This title covers the details of our work properly, by focusing on the quality of squid mantle cut treated with mint extract and high-pressure processing during refrigerated storage. Thus, we prefer to keep the present title.

2- Please, update the section of introduction with recent information from recent References (2022-2024), to make your study more interesting and more significant to the reader.

for ex. Coupling ozone with microbubbles (OMB) water for food disinfection: Effects on microbiological safety, physicochemical quality, and reducing pink discoloration of jumbo squid (Dosidicus gigas) – ScienceDirect

Evaluating the effects of nanoparticles combined ultrasonic-microwave thawing on water holding capacity, oxidation, and protein conformation in jumbo squid (Dosidicus gigas) mantles – ScienceDirect Animals | Free Full-Text | Innovative Seafood Preservation Technologies: Recent Developments (mdpi.com)

***Actually, the recently published information had been provided in the introduction (2021 – 2023: 5 papers), which were also related to the scope of present study. However, the first reference suggested by reviewer is related to the use of ozone-microbubble (OMB) water for disinfection of squid prior to refrigerated storage. The second one is related to the use of magnetic nanoparticles in combination with microwave and ultrasonic waves on the water holding capacity, oxidation of protein and lipid, and protein conformation, jumbo squid mantles. Those works are not related with our work, which focused on the use of HPP and natural extract on shelf-life of refrigerated squid mantle cut.

Since we are working on the textural improvement and quality of giant squid, we will cite those two references in our new manuscript, which will be submitted within 1-2 months. 

3- In the section 2.2. and 2.4 please try to edit these steps referring to their references sources in order to be clearer and more scientific. Please write these steps from the ref.

***The references were cited for all procedures used in the present study. The necessary details were provided and some important methods were described in details. The repeated general details from the former literatures should be avoided to reduce the plagiarism.

4- Please, provide a better figures of 2,3 and 4 of higher resolution, because the current figures is not clear

***Sorry for the poor presentation of those three figures. We found that the sizes of figures are small and the details in X and Y-axis are hard to follow. To tackle such as drawback, the size and resolution of all figures has been improved. Please see Figure 2, Figure 3, and Figure 4.

5. Comments on the Quality of English Language

Dear, this research needs a simple linguistic review, and it is written in good language, but it is better for it to be reviewed by a native English speaker.

***Thank you for the comment. The authors have cross-checked the manuscript. The software’ Grammarly’ has been used to avoid grammatical errors for the whole manuscript.

Reviewer 2 Report

Comments and Suggestions for Authors

Please provide an abbreviation list

Change the first letter of Gram-negative and Gram-negative from Uppercase to lower case in the middle of the sentences.

When citing a reference containing more than 2 authors cite it as the first name et al. instead of writing the names of all authors.

Figure 3. caption organization isn't the same as the charts. Please correct

It is denated that (a) is the pH while in the figure it is the TVB, (b) TBV  while it is TMA and (c) TMA while in the figure is the pH. 

The rest of comments are present as comments in the manuscript pdf file attached

Author Response

Response to reviewer

****Thank you so much for the insightful comment. All queries have been responded and the corrections have been made as highlighted in green color.

  1. Please provide an abbreviation list

*** Authors mostly used the abbreviation to represent the known quality indices (microbiological and chemical parameters). Furthermore, the abbreviations were given to the samples with different treatments in order to avoid the plagiarism. Nevertheless, the description was given for all abbreviations at the first appearance for better understanding.

Also, to make the figure and table easy to follow, the description of all abbreviation of samples were given. Please see all the figures captions and table footnote.

Therefore, authors prefer not to provide the list of abbreviation to avoid the repetition or redundance.

  1. Change the first letter of Gram-negative and Gram-negative from Uppercase to lower case in the middle of the sentences.

***Thank you very much for suggestions. The corrections have been made throughout the text. Please see line 226, 331, 340, 427, 616, and 621.

  1. When citing a reference containing more than 2 authors cite it as the first name et al. instead of writing the names of all authors.

***Authors strictly followed the rule of Reference List and Citations Style Guide for MDPI Journals.

  1. Figure 3. caption organization isn't the same as the charts. Please correct. It is denated that (a) is the pH while in the figure it is the TVB, (b) TBV while it is TMA and (c) TMA while in the figure is the pH.

***Sorry for the mistake. The corrections have been done as highlighted in green. Please see line 401 – 402.

  1. The rest of comments are present as comments in the manuscript pdf file attached

***Thank you very much for comments. The corrections have been made following the reviewer’s suggestions. Please see line 236, 303, and 336.

Reviewer 3 Report

Comments and Suggestions for Authors

The publication was prepared at a good substantive and technical level.

Detailed comments:

l. 22,25 - The M4P4 code is unnecessary here and should be removed.

l. 52 - Unnecessary dot after "However".

l. 86 (and next) - Naher, et al. [14] - Please verify the citation rules.

l. 107 - Agree on the number of significant digits.

l. 110 - Verify the dimensions of the pieces. It should be 5x5 cm or 25 cm2.

l. 110 - Is ME-80 the same as ME in line 87? Use the appropriate designations.

l. 120 - Why was this processing time chosen? The pressure is justified in line 48. The justification for the time is missing.

l. 172,177 - The equations are very obvious. They should be removed.

l. 181 - Please provide the attachment ID according to the SMS catalog.

l. 184-191 - Is there approval to conduct an experiment involving humans?

l. 204 - The dot after "Inc" is missing.

Figure 1. - Remove the background from the markers for better readability of the charts. The bars (Fig. 1b) are unreadable because they overlap. Should be corrected.

Figure 2,3,4. - Charts are not very readable (e.g., too small letters in captions, too narrow bars lose patterns). The captions should be explained here. Please change the way the information is presented.

l. 418 - First occurrence of the acronym TMAO. Please explain it.

Table 2. - Use capital letter at the beginning of word in the table header.

l. 665, 830, 861 - Veryfy the journal name.

Author Response

Response to reviewer

The publication was prepared at a good substantive and technical level.

***Thank you very much for your kind understanding. All your comments and suggestions have been responded and some corrections have been made in the manuscript as highlighted in yellow color.

Detailed comments:

  1. Line 22,25 - The M4P4 code is unnecessary here and should be removed.

***Thank you for suggestion. The code “M4P4” has been changed to “Sample” in line 22 and it has been removed from line 26.

  1. Line 52 - Unnecessary dot after "However".

***Dot (.) has been replaced with comma (,). Please see line 52.

  1. Line 86 (and next) - Naher, et al. [14] - Please verify the citation rules.

***This citation “Naher, Nilsuwan, Palamae, Hong, Zhang, Osako and Benjakul [14]” has been made according to Reference List and Citations Style Guide for MDPI Journals. The abbreviation “et al.” can be used when the article has authors more than 10.

  1. Line 107 - Agree on the number of significant digits.

***The correction has been done. Please see line 107.

  1. Line 110 - Verify the dimensions of the pieces. It should be 5x5 cm or 25 cm2.

***To avoid misunderstanding, the dimension of the pieces has been corrected with (5 ´ 5 cm). Thank you for the valuable suggestion. Please see line 110.

  1. Line 110 - Is ME-80 the same as ME in line 87? Use the appropriate designations.

***The “ME-80” has been replaced with “ME” for the consistency throughout the text. Please see line 87.

  1. Line 120 - Why was this processing time chosen? The pressure is justified in line 48. The justification for the time is missing.

***The time of high-pressure processing (HPP) has been selected (5 min) based on the result of “Inactivation of planktonic cells by HPP” as discussed in section 3.2. Please see line 245 – 248 and 261 – 262.

It was found that “lower survival of cells was noted for Pseudomonas aeruginosa (PA) and Shewanella spp. (SWS) after HPP at 400 MPa for 5 and 7 min, compared to those of PA-HPP and SWS-HPP treated for 1 and 3 min (p<0.05). No difference between counts of PA-HPP and SWS-HPP at 5 and 7 min was found (p>0.05). Overall, HPP treatment time had the potential influence on inactivation of cells. Therefore, HPP at 400 MPa for 5 min was selected for microbial inactivation.”.

  1. Line 172,177 - The equations are very obvious. They should be removed.

***Those equations along with their related descriptions have been removed from the text following the reviewer’s suggestions. Please see line 167 – 174. Thank you very much.

  1. Line 181 - Please provide the attachment ID according to the SMS catalog.

***More details have been provided. Please see line 177 – 178.

  1. Line 184-191 - Is there approval to conduct an experiment involving humans?

***The sensory evaluation was approved by the ethical committee, Prince of Songkla University.  The ethical number was PSU-HREC-2023-007-1-1. The details have been provided the manuscript. Please see line 189-191.

  1. Line 204 - The dot after "Inc" is missing.

***Dot has been provided. Please see line 204.

  1. Figure 1. - Remove the background from the markers for better readability of the charts. The bars (Fig. 1b) are unreadable because they overlap. Should be corrected.

*** The figure and symbols representing the different treatments have been improved and enlarged. It can be readable in the new version. Please see line 231. Thank you. 

  1. Figure 2,3,4. - Charts are not very readable (e.g., too small letters in captions, too narrow bars lose patterns). The captions should be explained here. Please change the way the information is presented.

***The resolution and size of the figures have been improved. Please see Figure 2, Figure 3, and Figure 4. Moreover, the captions have been provided for Figure 2, 3, and 4 according to the reviewer’s suggestions. Please see line 304 – 306, 398 – 400, and 549 – 550.

  1. Line 418 - First occurrence of the acronym TMAO. Please explain it.

*** This query is also mentioned by another reviewer. Full name of TMAO has been provided at the first time. Please see the first appearance of TMAO in line 336 as highlighted in green.

  1. Table 2. - Use capital letter at the beginning of word in the table header.

***The words in table header have been corrected. Please see Table 2.

  1. Line 665, 830, 861 - Verify the journal name.

***All journal names have been re-checked and corrected. Please see line 680, 845, and 876. Thank you for valuable suggestions.

Reviewer 4 Report

Comments and Suggestions for Authors

This study aimed to investigate the combined effect of mint extract (ME) and HPP on quality changes of squid mantle cut during refrigerated storage. The study is interesting for students, academics and general public; however, some comments should be attended.

Lines 105 and 116; please mention how many replicates were used for each treatment and control.

Line 200: A completely randomized design is not enough to statistically analyze the results. In the treatments the authors have the interaction ME extract and HPP, therefore I suggest to analyze the data with a factorial design with blocks; where factor A could be ME concentrations (0, 200, 400), factor B the HPP (200, 400) and storing days the blocks (0,3,6,9,12,15) because it is well known that the more storing days of SMC, the more effects on SMC could be observed in some variables.

Figure 2,3, 4 are difficult to read, use bigger figures please.

Table 2 and Figure 5: why day 12 in M4P4 if authors stated a storing period of 15 days?

Comments on the Quality of English Language

Minor editing of English language required

Author Response

Response to reviewer

This study aimed to investigate the combined effect of mint extract (ME) and HPP on quality changes of squid mantle cut during refrigerated storage. The study is interesting for students, academics and general public; however, some comments should be attended.

****Your understanding in our work is highly appreciated. Thank you so much for the invaluable comments and suggestions. All queries have been responded and the corrections have been made as highlighted in turquoise.

  1. Lines 105 and 116; please mention how many replicates were used for each treatment and control.

***Thank you for comments. All experiments were carried out in triplicate. This detail was mentioned in the text. Please see line 201 – 202.

  1. Line 200: A completely randomized design is not enough to statistically analyze the results. In the treatments the authors have the interaction ME extract and HPP, therefore I suggest to analyze the data with a factorial design with blocks; where factor A could be ME concentrations (0, 200, 400), factor B the HPP (200, 400) and storing days the blocks (0,3,6,9,12,15) because it is well known that the more storing days of SMC, the more effects on SMC could be observed in some variables.

***Thank you for your insightful suggestion. We decided to use Completely Randomized Design in our study since we want to find the best condition for extending the shelf life of squid ring treated with mint extract in combination with HPP. Therefore, we considered each treatment as the individual sample and make comparison at each storage time. For the same treatment, the function of time was also analyzed to elucidate the influence of storage time on quality of different samples. Overall, we could end up with the best treatment for shelf-life extension of squid ring, in which the goal of study was achieved.

  1. Figure 2,3, 4 are difficult to read, use bigger figures please.

***The resolution and size of all figures has been improved as per the reviewer’s suggestion. Please see Figure 2, Figure 3, and Figure 4. Please see line 304 – 306, 398 – 400, and 549 – 550.

  1. Table 2 and Figure 5: why day 12 in M4P4 if authors stated a storing period of 15 days?

***It is true that the total period of storage was 15 days. However, M4P4 had total viable count (TVC) higher than 6 log CFU/g after 15 days. This count is not acceptable for human consumption. Therefore, M4P4 at 12 days, which was the longest storage time, while TVC was still under the limit, was selected for sensory evaluation and next-generation sequencing (NGS). This detail had been mentioned in text. Please see line 185 – 187.
